# A Meta-Omics Analysis Unveils the Shift in Microbial Community Structures and Metabolomics Profiles in Mangrove Sediments Treated with a Selective Actinobacterial Isolation Procedure

**DOI:** 10.3390/molecules26237332

**Published:** 2021-12-02

**Authors:** Miguel David Marfil-Santana, Anahí Martínez-Cárdenas, Analuisa Ruíz-Hernández, Mario Vidal-Torres, Norma Angélica Márquez-Velázquez, Mario Figueroa, Alejandra Prieto-Davó

**Affiliations:** 1Unidad de Química-Sisal, Facultad de Química, Universidad Nacional Autónoma de México, Sisal 97356, Mexico; miguelmarfil81@gmail.com (M.D.M.-S.); aluisarh@hotmail.com (A.R.-H.); mario-fate@hotmail.com (M.V.-T.); angelica.marvel@quimica.unam.mx (N.A.M.-V.); 2Facultad de Química, Universidad Nacional Autónoma de México, Ciudad de México 04510, Mexico; amartinez@quimica.unam.mx (A.M.-C.); mafiguer@unam.mx (M.F.)

**Keywords:** mangrove sediments, PKS, NRPS, metagenomics, bioactive natural products, metabolomics

## Abstract

Mangrove sediment ecosystems in the coastal areas of the Yucatan peninsula are unique environments, influenced by their karstic origin and connection with the world’s largest underground river. The microbial communities residing in these sediments are influenced by the presence of mangrove roots and the trading chemistry for communication between sediment bacteria and plant roots can be targeted for secondary metabolite research. To explore the secondary metabolite production potential of microbial community members in mangrove sediments at the “El Palmar” natural reserve in Sisal, Yucatan, a combined meta-omics approach was applied. The effects of a cultivation medium reported to select for actinomycetes within mangrove sediments’ microbial communities was also analyzed. The metabolome of the microbial communities was analyzed by high-resolution liquid chromatography-tandem mass spectrometry, and molecular networking analysis was used to investigate if known natural products and their variants were present. Metagenomic results suggest that the sediments from “El Palmar” harbor a stable bacterial community independently of their distance from mangrove tree roots. An unexpected decrease in the observed abundance of actinomycetes present in the communities occurred when an antibiotic-amended medium considered to be actinomycete-selective was applied for a 30-day period. However, the use of this antibiotic-amended medium also enhanced production of secondary metabolites within the microbial community present relative to the water control, suggesting the treatment selected for antibiotic-resistant bacteria capable of producing a higher number of secondary metabolites. Secondary metabolite mining of “El Palmar” microbial community metagenomes identified polyketide synthase and non-ribosomal peptide synthetases’ biosynthetic genes in all analyzed metagenomes. The presence of these genes correlated with the annotation of several secondary metabolites from the Global Natural Product Social Molecular Networking database. These results highlight the biotechnological potential of the microbial communities from “El Palmar”, and show the impact selective media had on the composition of communities of actinobacteria.

## 1. Introduction

Mangroves are coastal estuarine ecosystems covering 60–70% of the coastlines in the tropics and subtropics [1], and they have an estimated economic value of 2000 to 9000 USD per hectare per year [2]. Mangroves act as barriers against erosion by the wind and ocean waves, and create an important buffer zone between the land and ocean [3,4]. These tidal forests are highly productive environments with robust microbial communities, which play a crucial role in supporting the local food chain. As in terrestrial forests, bacteria in mangrove sediments play essential roles in benthic food webs as remineralizers of organic detritus and recyclers of essential nutrients [5], and their actions impact marine life at all trophic levels [6,7,8]. With approximately 775,555 Ha of mangroves, Mexico hosts one of the largest mangrove regions in the world [9]. “El Palmar” is a protected RAMSAR site in Mexico covering c.a. 50,200 Ha along the Yucatan coast. The area is characterized by karstic sediments, and these unique sediments are found throughout the coastal wetlands of the peninsula [10]. Mangrove sediment ecosystems host a diverse and complex microbial community, including members of the phyla *Firmicutes, Actinobacteria,* and *Proteobacteria*, with families such as *Chromatiaceae*, *Rhodospirilaceae*, *Chloroflexaceae* [11,12,13]. Historically, studies of the microbial communities within mangroves have focused on describing the diversity and structure of the communities, as well as their roles in carbon and nitrogen geochemical cycles [14,15,16,17,18,19]. These microbial communities are also likely to be a source of novel enzymes and small molecules of industrial and pharmaceutical interest [20]. Secondary metabolites produced by microbes isolated from mangrove sediments have yielded several compounds with pharmaceutical and industrial relevance, such as lumazine peptide aspergilumamide A, pyrrolizidine alkaloid penibruguieramine A, and the potential anticancer peptides pullularins E and F [21,22,23]. Furthermore, enzymes valuable to industry have been identified, such as lipase A, a GH44 family endoglucanase, and the salt-tolerant endo-β-1-4-glucanase Cel5A [21,22,23,24,25,26]. Selective isolation techniques have been used to culture microorganisms from sediments, mainly actinomycetes, which produce desired secondary metabolites. For example, a *Streptomyces* strain isolated from the top sediment layer of a mangrove in Koringa, India, possessed antimicrobial activity against *Candida albicans* and *Pectinotrichum llanense* [27], while another *Streptomyces* strain isolated from Indian mangrove sediments was reported to possess promising antioxidant, antimicrobial, and anti-inflammatory agents [28]. However, descriptions of the biotechnological potential of mangrove microbial communities through the use of genome mining is scarce, and little research has focused on identifying genes involved in the biosynthesis of medically and industrially relevant compounds from these sediments [29,30,31,32]. Among these genes, polyketide synthase (PKS) and non-ribosomal peptide synthetase (NRPS) genes are the most commonly used to assess the natural product potential of a community due to their involvement in the biosynthesis of numerous bioactive compounds [33,34,35].

Environmental factors have been suggested as the main regulators of complex taxonomic structures in microbial communities from mangrove sediments [36]. The unique environmental characteristics of karstic sediments found in the Yucatan peninsula could therefore lead to novel microbial communities with high biotechnological potential and address questions about the biosynthetic capacities of communities within these highly adaptive and productive ecosystems [11,14,17,20]. This work describes a multi-omics approach to study the “El Palmar” mangrove sediment microbial communities and their biotechnological potential. Within these sediments, the approach provides an opportunity to evaluate the community structure, as well as identify genes involved in environmentally relevant biosynthetic pathways, including the biosynthesis of polyketides and non-ribosomal peptides. Furthermore, this study uses metagenomics to analyze changes in the microbial community structure and its metabolome following the microbial community’s exposure to a selective medium commonly used to enrich for actinomycetes. Although other studies have used these approaches to explore biosynthetic capabilities of microbes for bioremediation [37] to our knowledge, this is the first bioprospecting study where multi-omics has been employed to analyze the performance of selective media as a means to enhance secondary metabolite discovery.

## 2. Results

### 2.1. Sampling Location, Collection Dates and Experimental Treatments

Three samples from mangrove sediments were obtained on January 2017 (21.121251 N, −90.074685 W) inside “El Palmar” natural reserve. These samples were taken at the same site, but at different distances from mangrove tree roots. MR samples are sediments that were collected directly over roots; MA samples are sediments collected close to roots but not directly over them (approximately 1.5 m away); anrd MC samples are sediments collected in a nearby water channel with a constant flow of water and not obviously influenced by mangrove roots (Appendix A). To understand the effects of using an actinomycete-selective medium on the microbial community from mangrove sediments, a second sampling was performed on December 2018 (same season, but roughly two years apart). For this medium-enrichment metagenomic experiment, only sediments from the mangrove water channel (MC) were sampled, mainly due to: (1) Metagenomes from the three previous samples showed no significant differences in their actinomycete community distribution (see below), and (2) A parallel cultivation-dependent study showed a greater number of bioactive strains (via “in-house” bioassays) from these sediment samples (data not shown). The 2018 samples (triplicate samples from the mangrove channel (MC)) were labeled as T0 for the medium-enrichment experiment. A control for the enrichment conditions, including sediments in a flask for 30 days, as well as autoclaved mangrove water, but with no medium or antibiotics added was labeled TA. Medium-enriched sediments without and with antibiotics in this experiment were labeled as TM and TMa, respectively (see methodology).

### 2.2. Environmental Parameters at Location

Water parameters at sediment depth for the January 2017 samples were: temperature 22.3 °C, 7.4 mg/L of dissolved O_2_, salinity of 3.49, pH of 8.3, and a 31 mV for redox potential. The water from the December 2018 sample at the MC (mangrove channel) area had the following water parameters at sediment depth: temperature 22.6 °C, 7.7 mg/L of dissolved O_2_, salinity of 3.48, pH of 8.25 and 37 mV for redox potential (Appendix A).

### 2.3. Metagenomic Analyses

#### 2.3.1. Sequence Processing and Filtering

High throughput sequencing of samples corresponding to MR, MA, MC, T0, TA, TM, and TMa generated 6 to 12.9 GB of pair end information. Metagenomes had varying number of reads, e.g., MC had 162,118,455, while TM had 6,953,193 (Appendix A). Pair-end assembled metagenomes were uploaded and processed on the open platform MG-RAST, where automated taxonomic and functional gene identification was performed.

The highest number of sequences that passed the quality control (QC) filter was 95.91% and belonged to the TA metagenome, while only 24.28% of the MA metagenome sequences passed this filter (Appendix A). However, due to the high number of sequences in the MA raw metagenome (39,014,588), the 24.8% value still resulted in a considerable amount of sequences for use in the study (Appendix A). Sequences that passed the QC were automatically annotated as either proteins with or without known functions. The TMa metagenome had the highest percentage of proteins with known function (72.86%), while MA had the lowest percentage (27.16%) (Table 1). The highest number of 16S rRNA sequences was observed in the MA metagenome, with 3.22% of the metagenome being labeled as 16S genes. Only 0.29% of sequences of the TMa metagenome were labeled as 16S genes (Table 1).

#### 2.3.2. Taxonomic Distribution within Metagenomes from Mangroves

Taxonomic annotation of metagenomes in MG-RAST was done by comparison to the NCBI RefSeq database (http://www.ncbi.nlm.nih.gov/RefSeq/; accessed on 21 August 2019), which assigns taxonomy to the functional genes recovered. Rarefaction curves show that sequencing efforts for all mangrove metagenomes were sufficient to guarantee a good representation of the microbial community composition present in all mangrove sediment samples (Appendix A).

α-diversity estimates at the phylum level were similar for all metagenomes of samples collected in 2017 (MR, MA, MC) and the one collected in December 2018 (T0). However, richness significantly decreased when enrichment experiments were applied (Figure 1). When the microbial community composition was analyzed, the three mangrove microbial communities sampled in January 2017 did not differ significantly from one another, however, there were clear differences when compared to the sample recovered in December 2018 and to the experimental metagenomes (Figure 1). At the same time, α -diversity did not differ between the MC sediments from the second collection date (T0) and the water control sediments constrained to a 2 L flask with mangrove water for 30 days (TA) (Figure 1). However, a clear reduction in the α-diversity was observed in the 2018 MC sediments when nutrient rich media and antibiotics were added to the mangrove water used in the enrichment experiment with the sediments, and the communities were grown for 30 days in a 2 L flask (T0, TA vs. TM, TMa) (Figure 1).

### 2.4. “El Palmar” Mangrove Sediment Metagenome

Domain level analysis of the metagenomes shows that Bacteria and Archaea are the dominant microorganisms in this environment, and that the enrichment experiment reduced the number of the latter (Appendix A). Interestingly, a higher number of viral sequences were observed in the metagenome where nutrient-rich medium was amended with antibiotics (TMa) (Appendix A).

With regard to the Archaea, the sediments of “El Palmar” were dominated by the domain Euryarchaeota, which contributed more than 80% of the identified sequences (Appendix A). Sediments from the December 2018 sample (T0) show a significant abundance of Thaumarchaeota and Crenarchaeota (Appendix A), while the addition of a nutrient-rich medium and antibiotics decreases their abundance (Appendix A). All metagenomes show low abundances of Korarchaeota and Nanoarchaeota (Appendix A). Differences identified in the archaeal communities were highlighted in a principal component analysis (PCA), which compared the two time points at which the environmental and experimental communities were sampled (Figure 2).

For Bacteria, “El Palmar” mangrove sediments were highly dominated by Proteobacteria, with this phylum representing between 49–60% of the sequences (Appendix A). Compared to the January 2017 sediments, the December 2018 sediments showed significantly higher abundance of this phylum and of Actinobacteria, while there was a significant reduction in the abundance of Cyanobacteria and Bacteroidetes (Appendix A). When the sediments were incubated in the lab for 30 days (T0 vs. TA), a significant increase in the abundance of Proteobacteria was observed, while Firmicutes abundance was significantly lower (Appendix A). When the sediments were enriched with a high-nutrient medium, abundances were also observed to change (T0 vs. TM) (Appendix A). Proteobacteria and Actinobacteria decreased significantly in abundance over the time of the experiment with and without antibiotics added to the medium (Appendix A). On the other hand, growth on nutrient-rich medium also had an effect on the abundance of Firmicutes and Bacteroidetes, increasing them significantly (Appendix A). When antibiotics were added to the medium (TMa), Bacteroidetes abundance increased, matching that of Proteobacteria, but Firmicutes abundance decreased again (Appendix A). Higher taxonomic level analyses were performed to further depict changes in bacterial abundances. These changes show a significant decrease in the class Actinomycetia in sediments amended with nutrient-rich medium with and without antibiotics (from 5–6% in the environment communities to <2% in the amended media, TM and TMa) (Figure 3). Significant changes between communities from the environment (MA, MR, MC and T0) and experimental communities grown in nutrient-rich medium and amended with antibiotics (TM and TMa) were also observed in Clostridia, Cyanobacteria, Planctomycetia, Bacilli, Flavobacteria, Sphingobacteriia, Bacteroidia, Cytophagia, and Deferribacteres (Appendix A).

Changes in bacterial populations were also evident in abundance heat maps (Appendix A), and coincide with principal component analyses (PCA). No statistically significant differences (*p* > 0.05) were observed between biannual samples, while experimental microbial communities from sediments grown for 30 days with nutrient-rich medium with and without antibiotics separated into their own groups (Figure 4). No significant differences were observed between microbial communities from the 2018 environmental sample (T0) and the sediment control incubated in autoclaved water and maintained under laboratory conditions (TA). Order level taxonomy analysis, however, reveals a difference between sediment microbial communities sampled in January 2017 and in December 2018, suggesting that higher resolution analyses could give further insights into microbial community structure changes over time (Appendix A).

Due to the frequent use of high-nutrient and antibiotic-amended media for actinomycetes cultivation, an analysis of the bacterial composition of families within the order Actinomycetales was highly relevant. This analysis revealed that, at the Family level, abundances of actinobacterial communities remained constant in sediment samples independent of their contact with mangrove sediments (MR, MA, MC samples from 2017) and at different time points (MC vs. T0), but changed significantly (*p* < 0.05) when exposed for 30 days to an actinomycete-selective medium. In total, 12 families showed similar abundance distributions in the five analyzed metagenomes including: Streptomycetaceae, *Mycobacteriaceae*, *Frankiaceae,* and *Pseudonocardiaceae* (Appendix A). However, when sediments were amended with nutrient-rich medium and antibiotics, the abundance of *Nocardiacea* was significantly increased and a decrease in the abundance of *Frankiaceae*, *Pseudonocardiaceae,* and *Mycobacteriaceae* was observed (Appendix A).

### 2.5. Functional Distribution in Metagenomes

All functional assignments were performed using the M5rna database from MG-RAST. All seven metagenomes show similar abundance distribution of genes involved in primary metabolism (Figure 5). However, statistical comparisons performed using STAMP (https://beikolab.cs.dal.ca/software/STAMP; accessed on 21 August 2019) showed significant differences (*p* < 0.05) among the metabolism genes identified within the communities grown with and without nutrient-rich medium and antibiotics (MR, MA, MC, T0, TA, vs. TM and TMa). Abundances of genes involved in iron acquisition metabolism, nitrogen metabolism, sulfur metabolism, motility and chemotaxis metabolism, phages, prophages metabolism, respiration metabolism, and transposable elements metabolism all showed changes when environmental communities were subjected to cultivation conditions (Figure 6 and Appendix A). When the metabolism genes of individual metagenomes are studied closer, differences in gene abundances can be observed. For example, in the experimental metagenomes (TM and TMa), the sulfur metabolism shows decreasing abundances of inorganic and organic sulfur assimilation genes (e.g., alkenosulfonates-binding protein, DMSP demethylase, and sulfate permeases and sulfur oxidation proteins). Furthermore, in these metagenomes, genes associated with iron metabolism increase, including genes related to siderophores and transporters (Figure 6, Appendix A). Moreover, genes related to nitrogen metabolism differ between the environmental metagenomes from the two time points (MR, MA, MC vs. T0), where a decrease in nitrogen fixation genes and an increase in denitrification and nitric oxide synthesis can be observed in the 2018 environmental metagenome. Differences between 2018 environmental and experimental metagenomes (T0 and TA vs. TM and TMa) can also be observed in genes related to nitrate and nitrite ammonification, nitrogen fixation, dissimilatory nitrite reductase, nitric oxide synthesis, and ammonia assimilation (Appendix A). Lastly, genes associated with phosphate metabolism are more abundant in the 2017 metagenomes, including genes associated with phosphate uptake found in cyanobacteria (Appendix A). The only increase in metabolism genes observed in the experimental metagenomes (TM and TMa) was with the alkylphosphonate utilization genes.

A closer look at genes associated with secondary metabolism, which is the focus for natural product discovery, revealed clear differences between the environmental microbial communities at the two time points (MR, MA, MC vs. T0), and between the environmental microbial communities, the experimental metagenomes, and the control (Figure 7). The main differences were observed among genes related to plant hormones, biologically active compounds in metazoan cell defense and differentiation, plant alkaloids, biosynthesis of phenylpropanoids and bacterial cytostatics, differentiation factor, and antibiotics (Appendix A). There was a decrease in the abundance of genes related to plant alkaloids after 30-day incubation of sediments and mangrove water (T0 vs. TA) (Appendix A). On the other hand, there was an increase in the abundance of genes related to the biosynthesis of phenylpropanoids and plant hormones in the 2018 environmental metagenome as compared to the 2017 metagenomes (T0 vs. MC, MA, MR), which remained high even in the nutrient-rich (TM) and antibiotic amended (TMa) metagenomes (Appendix A). Genes related to dormancy and sporulation also show changes among metagenomes, with increases among all sporulation categories in the experimental metagenomes amended with nutrients and antibiotics (TM and TMa) (Appendix A), including the increase of genes related to biofilm activated persister cells.

A comparison of phage-related genes showed an increase of sr1t-like streptococcal phage genes in the two experimental metagenomes (TM and TMa), as well as phage integration and excision genes (Appendix A).

### 2.6. Metagenome Processing

Polyketide synthases (PKS) and non-ribosomal peptide synthetases (NRPS) were mined from raw metagenome sequences using signature motifs for common PKS genes, such as the ketosynthase (KS) domain type 1, and those affiliated with NRPS synthesis such as condensation (C) domains and adenylation (A) domains. For all metagenomes, 27,390 C domains and 684 KS domains were identified; the signature motifs of the NRPS-A domains were non-specific and therefore not analyzed (data not shown) (Appendix A). KS, C, and A domain sequences were also mined from assembled metagenomes by performing BLAST searches against the NaPDoS [38], ESNaPD [39], and SBSPKS [40] databases (Table 2; Appendix A). Further, the resulting regular expressions for KS and C domains were confirmed by performing a Blast search against these databases which drastically decreased the number of identified sequences (Table 2). These analyses showed that enrichment experiments affected the number of sequences related to natural product domains that could be identified in the metagenomes. When the potential of PKS and NRPS production from microbial communities present in environmental and experimental metagenomes was assessed by adding the number of domains found using all methods, all metagenomes showed similar biosynthetic potential (Table 2). However, sediments from 2017 (MR, MA, and MC) showed overall higher abundance of these sequences than the 2018 sediment (T0), and the lowest secondary metabolite production potential was observed in the metagenome amended with nutrients and antibiotics (TMa) (Table 2).

### 2.7. AntiSMASH

Processed and assembled metagenome contigs (>2000 bp) (Trimmomatic V0.39, MEGAHIT-1.2.9) were used to mine for bacterial secondary metabolite clusters using antiSMASH (Table 2 and Appendix A), and the trend observed with BLAST searches was confirmed, i.e., the 2017 sediments showed a higher number of potential biosynthetic pathways (644) than the 2018 sediments (407). However, the sediments amended with high nutrients did not show a decrease in the number of possible secondary metabolite clusters observed (Table 2). Some domains found by BLAST metagenome mining were also observed by antiSMASH (shared) (Table 2 and Appendix A, Appendix A). Pathways observed with antiSMASH belonged to compounds from categories such as NRPS, PKS, terpenes, phosphonate, ectoine, lantipeptide, sactipeptide, bacteriocin, cyanobactin and lassopeptide (Appendix A, Figure 8). Identification of genes with similarities to reported compounds in databases were found using both mining methodologies and included genes similar to: nostopeptolide, jamaicamide, stigmatellin, bacillibactin, fengycin, pyochelin, bacitracin, yersiniabactin, mycosubtilin, tyrocidine, fengycin, sporolide, polyunsaturated fatty acid, alkylresorcinol, omega-3-fatty acid, lovastatin, rifamycin, curacin, lipomycin, stigmatellin, myxothiazol, erythronolide, epothilone, and virginiamycin.

### 2.8. Metabolomics of Microbial Communities’ Enrichments

To explore the chemical diversity and specific metabolites produced by the microbial communities (TA, TM, and TMa) under various growth conditions, we performed feature-based molecular networking (MN) and untargeted metabolomics analyses of the detected metabolites. For the MN construction, the Global Natural Product Social Molecular Networking (GNPS) platform [41] assembles MS/MS spectra by similarity networks (similar structures present comparable fragmentation patterns), and metabolic clusters are formed by nodes that represent precursor ions. A comprehensive MN was generated for spectra with a minimum of four fragment ions and grouped the metabolite features into 60 chemical families (>3 nodes), 98 families of two metabolite features, and 984 singletons (Figure 9). It is important to mention that the number of nodes do not match to the number of metabolites, as different adducts or different charge states of the same chemical species can generate different nodes. Thus, the MN provides an overview of the chemical diversity of the microbial communities. The most common chemical superclass observed was lipids and lipid-like molecules (commonly found in sediment or soil samples) and organoheterocyclic compounds. Notably, an important number of no hits were found.

From the GNPS analysis, only three compounds were annotated: actiphenol and secocycloheximide A produced by *Streptomyces* spp., and lumichrome produced by *Micromonospora* sp. [41]. In addition, using untargeted metabolomics based on the putative metabolites produced by the communities and the information from the metagenome analysis, we tentatively identified a series of bacteriocins and soraphens, as well as tetronomycin across all samples (Table 3 and Figure 10). Interestingly, some metabolites were only found in the communities grown under enrichment treatments (Table 3 and Figure 10).

## 3. Materials and Methods

### 3.1. Sample Collection

Three sediment samples, 500 g each, were collected from the natural reserve “El Palmar” located in the Yucatan peninsula, Mexico (21.121251 N, −90.074685 W, Appendix A), using a homemade coring device with an acrylic tube measuring 10 cm in diameter and 50 cm long, on January 2017. Each sample represented a different microenvironment of the mangrove: Mangrove roots (MR), sediments that were directly on top of mangrove tree roots; in between mangrove tree area (MA), sediments that were collected close to mangrove roots but not directly above them (approximately 1.5 m away); and water channel (MC), sediments collected away from direct mangrove root influence and on a permanent water flowing channel (Appendix A). After a first metagenomic analysis and further cultivation efforts (data not shown), a second sampling was performed in December 2018, where only sediments from the water channel area (MC) were obtained for medium-enrichment studies in the lab, during this sampling time, three cores were used as triplicate for the experiment. Regardless of dates obtained, all sampling was done in triplicate with the same coring device that went down to 50 cm below top sediments under approximately 60 cm of water. All samples were kept in sterile Whirl-Pack bags and transported to the lab in a cooler. All samples were kept at 4 °C for 24 h before processing. Environmental parameters of the water above the sediments were obtained using a YSI multiparametric device (Xylem Inc., Rye Brook, NY, USA).

### 3.2. Sediment Enrichment Experiments to Investigate the Effect of an Actinomycete-Selective Medium and Addition of Antibiotics on the Microbial Community Structure of Mangrove Sediments

A total 250 g of sediments from the December 2018 samples were inoculated into 1L flasks with three different enrichment treatments by triplicate: (1) Water control (TA), with 1 L mangrove water that was filtered through a mesh to remove large particles and sterilized by autoclaving, (2) A1 Medium Control (TM), where the sediment was amended with 1 L of A1 medium (1 L mangrove filtered water, 2 g peptone, 10 g starch, 4 g yeast extract), (3) A1 Medium + antibiotics (TMa), where the sediment was amended with 1 L of A1 medium (prepared exactly as described above), plus, cycloheximide 10 µg/mL, gentamycin 10 µg/mL, and rifampicin 1 µg/mL. All enrichments were grown at 25 °C with 106 rpm shaking for 40 days. After the incubation period, the water phase was separated from the sediments and subjected to organic extraction. The remaining sediments were submitted to extraction of metagenomic DNA. DNA extractions from flask triplicates were standardized to same concentration and pooled together in order to send one pooled DNA sample per treatment.

### 3.3. Metagenomic DNA Extraction and Sequencing

Environmental DNA from sediments from the mangrove and DNA from triplicate enrichment experiments were extracted using the Quick-DNA^TM^ Fecal/Soil Microbe Miniprep (Zymo Research, Irvin, CA, USA) according to manufacturer’s instructions. DNA quality was assessed using a 1% agarose gel and quantified using a Nanodrop One (Thermo Scientific, Waltham, MA, USA).

DNA extractions were performed by triplicate and pooled by sampling zone or enrichment flask [42,43,44]. A total of 4 µg of DNA from each pool were sent for sequencing using Illumina HiSeq2500 2 × 150 pb (RTL-GENOMICS, Lubbock, TX, USA).

### 3.4. Taxonomic and Functional Annotation

Unassembled sequences were uploaded to the MG-RAST server [45] under the accession numbers: [mgm4798281.3] (MR), [4763253.3] (MC), [mgm4763252.3] (MA), [mgm4865010.3] (T0), [mgm4865011.3] (TA), [mgm4865012.3] (TM), [mgm4865013.3] TMa. The M5r database (non-redundant protein database in MG-RAST) was used for metagenome annotation and comparisons [46] using the following parameters: e-value 10^−5^, minimum ID value 60%, and minimum alignment length 15 amino acids.

### 3.5. Statistical Analyses

All functional annotations were downloaded and normalized to sequence number to compare between metagenomes. Stack bar graphs were created with Prism 8^MT^ (GraphPad Software, San Diego, CA, USA) and manually modified. All taxonomic and functional annotations from MG-RAST were exported to the statistical analysis software for metagenomic profiles (STAMP v2+; https://beikolab.cs.dal.ca/software/STAMP; accessed on 21 August 2019) [47] for further statistical comparisons under the following parameters: CI method (DP: Asymptotic-CC, 0.95%); Statistical test (G-test (w/Yates) + Fisher’s; *p*-value filter < 0.05 and Difference between proportions (effect size < 1.0). All PCA analyses were performed using an ANOVA Tukey-Kramer (CI 95%) test; Two-side Fishers test; with *p*-value filter > 0.050 and effect size < 2.

### 3.6. Manual Curation of Metagenomic Data (Metagenomic Mining)

#### 3.6.1. Data Assembly

Sequences obtained from each pool were processed to work with high quality reads by trimming and filtering with Trimmomatic [48] using the following parameters: Illumina TruSeq3 to trim adaptors, LEANDING and TRAILENG to trim end point base pairs with Phred values under 25, SLIDINGWINDOW:2:25 and MINLEN:40 to eliminate reads with less than 40 bp.

After trimming, sequences were assembled with MEGAHIT [49], using the following parameters: preset meta-sensitive and min-conting 500. Quality of the assemblies was assessed using BBmap (https://sourceforge.net/projects/bbmap/; accessed on 21 August 2019) with bbwrap.sh, pileup.sh and QUAST (V5.0.2) with default parameters.

In-house pipeline for polyketide synthase (PKS) and non-ribosomal peptide domain (NRPS) search.

Contigs in all metagenomes were screened for condensation (C) domains of NRPS (C-NRPS) and ketosynthase domain (KS) of PKS (KS-PKS) with “regular expressions” designed by aligning 189 C-NRPS and 117 KS-PKS protein sequences obtained from the NaPDoS database (Appendix A). Aligned sequences were used to find the conserved motifs GPXXXXXTACSS (KS-PKS) and HXXXDG (C-NRPS) that belong to the catalytic centers of their respective domains. Common variations and substitutions in the amino acids within these motifs were recognized near the substrate binding site. The resulting regular expressions were annotated as follows: “GP[C,S,Q,A]{5}TA[C,S,Q,Y][S,T][S,A]” for KS-PKS and HH[A-Z]{3}DG for C-NRPS.

#### 3.6.2. Blast Searches on Contigs from Metagenomes

Blast searches for sequences with KS domains for PKS, and adenylation (A) and condensation (C) domains for NRPS were performed using the NaPDoS, eSNAP, and SBSPKS databases. Blast+ was used with the following parameters: query_gencode 1; num_alignments 1; max_hsps 1; outfmt 6.

#### 3.6.3. AntiSMASH Search on Contigs from Metagenomes

Contigs from all metagenomes with more than 1000 bp were screened for biosynthetic gene clusters (BCGs) involved in the synthesis of antibiotics and secondary metabolites using antiSMASH 4.1.0 [50].

Organic extraction of the filtrates from sediment enrichment assays were extracted with 300 mL of EtOAc for 16 h at 27 °C and 106 rpm shaking. The organic solvent was separated in a separatory funnel and dried with added NaSO_4_. The extract was then evaporated under pressure with a Buchi R-215 Rotovap and analyzed by LC-MS.

### 3.7. Untargeted Metabolomics

#### 3.7.1. Metabolite Extraction

The filtrates from all sediment’s cultures (TA, TM, and TMa) were extracted with 300 mL of EtOAC for 16 h at 27 °C in a rotatory shaker (106 rpm). The organic layers were separated dried with NaSO_4_, and under vacuum. Dry extracts were kept at 4 °C until LC-MS analysis.

#### 3.7.2. LC-MS/MS Analysis

The organic extracts from samples TA, TM, and TMa were dissolved in LCMS grade methanol at 3 mg/mL and filtered with a 0.22 μm membrane. Ultra-high-performance liquid chromatography (UPLC) was conducted on an Acquity H-Class instrument (Waters Inc.) equipped with a PDA (UV_max_ λ 190–500 nm) detector, using a 1.7 μm Acquity BEH Shield C_18_ 50 mm 2.1 mm column at 40 °C. The mobile phase (flow rate of 0.3 mL/min) consisted of a linear gradient between CH_3_CN-0.1% aqueous formic acid, from 15% to 100% of CH_3_CN over 8 min, then held for 1.5 min at 100% CH_3_CN, and returning to the starting conditions. High-resolution mass spectrometry (HRMS) analysis and MS/MS spectra were recorded on a Q Exactive Plus (ThermoFisher Scientific) mass spectrometer equipped with an electrospray ionization (ESI) source (acquired in positive and negative modes) at a full scan range (*m*/*z* 150−2000), with the following Instrument parameters: capillary voltage, 5 V; capillary temperature, 300 °C; tube lens offset, 35 V; spray voltage, 3.80 kV; sheath and auxiliary gas flow, 30 and 20 arbitrary units.

#### 3.7.3. Data Analysis

Raw MS/MS data from samples and solvents (blank) were converted to .mzML file format and uploaded to the Global Natural Products Social Molecular Networking (GNPS) server (http://gnps.ucsd.edu; accessed on 21 May 2021). MN was performed using the reference GNPS data analysis workflow [41]. For the molecular networking (MN) analysis, a parent mass and fragment ion tolerance of 0.01 and 0.02 Da were considered. For edges construction in the MNs, a cosine score over 0.70 was fitted, a minimum of four matching ions, two nodes at least in the top 10 cosine scores, and 100 of maximum connected components, were considered for the analysis. The MolNetEnhancer GNPS tool was employed for chemical classification and the score represents what percentage of nodes within a molecular family are attributed to a given chemical class [51]. GNPS spectral libraries and graphic visualization of the MNs were analyzed in Cytoscape 3.8.1 [52]. Finally, manually dereplication of the observed ions in the main peaks of the LC-MS/MS chromatograms was achieved by comparison of the UV-absorption maxima and HRMS-MS/MS data against bacterial secondary metabolites from the Dictionary of Natural Products 29.1, Dictionary of Marine Natural Products 2019, and SciFinder databases. All annotations are at confidence level 2–3 according to the metabolomics standards initiative and exact mass accuracy < 5 ppm [53].

## 4. Discussion

“El Palmar” is one of the Yucatan’s coastal estuarine systems where mangrove sediment bacteria play important roles as remineralizers of organic matter. Furthermore, since exploration of novel, non-traditional environments has previously yielded novel natural products [54], the unique karstic conditions of Yucatan’s mangrove sediments offers a potential new source of bacteria capable of producing novel secondary metabolites. To understand the microbial community composition of “El Palmar” mangrove sediments, main biogeochemical roles played by community members, and their biotechnological potential, a metagenomic study was performed. Traditional bioprospecting strategies used in the discovery of novel natural products include targeted cultivation of sediment bacteria, mainly actinomycetes, by using selective, nutrient-rich medium to improve their recovery [55]. However, cultivation independent evaluation of the effectiveness of these methods is lacking. In this study, the efficacy of a nutrient-rich selective medium to enhance the recovery of actinomycetes from “El Palmar” mangrove sediments was tested by comparing the native microbial community composition and its functional characteristics for secondary metabolism to the microbial community of medium-enriched sediments.

The bacterial community structure profiles of sediments from the “El Palmar” mangrove showed no significant differences in the overall bacterial abundances between the three samples obtained in 2017 and the one sample obtained in 2018. The results suggest most bacterial phyla in these communities were stable in the environment throughout a two year period, probably due to the site’s constant coverage with water during this season and close proximity to stable vegetation [56]. Significant differences between four phyla, including Actinobacteria, were observed, which could be a result of either different environmental conditions between time points or due to a batch effect during sequencing of the samples. However, this difference was not observed at the family level within the Order *Actinomycetales* (Appendix A), suggesting analyses that include selection of actinomycetes should not be affected by the differences in samples. Conversely, the archaeal community showed differences in the abundances of *Euryarchaeota*, *Thaumarchaeota*, and *Crenarchaeota*, suggesting that archaeal community structure may be controlled by different parameters than those controlling bacterial communities. Further, bacterial and archaeal communities have been shown to adapt differently to environmental changes and to react to different stimuli in the environment [57,58]. For the experiments on the effect of actinomycete-selective cultivation medium on microbial community structure, significant differences (*p* < 0.05) in bacterial, archaeal, and viral populations were observed between the environmental and the experimental microbial communities (no significant difference was observed with the autoclaved water control). After a month of sediment exposure to high-nutrient medium, an unexpected decrease in the abundance of the targeted microorganisms (class *Actiomycetia* (previously *Actinobacteria*), Order *Actinobacteria* and Family *Actinomycetales* (actinomycetes)) was observed. The A1 medium used (see methods) selected for only some families within the class, and did not appear to allow for a wider diversity of actinomycetes to thrive as a result of decreasing competition by bacterial classes susceptible to the antibiotic. This result was confirmed by parallel cultivation- dependent studies from these sediments where most isolates belonged to the *Streptomycetacea* and *Nocardiaceae* families (data not shown). Therefore, it is recommended that experiments employing selective media use multiple enrichment media to provide diverse conditions for different families within this class to thrive. The decrease in the number of actinomycete families was accompanied by a decrease in the total number of sequences from actinomycetes in the metagenome. The numbers went from approximately 6% in environmental metagenomes, to less than 2% in those from enrichment experiments. Specifically, the order *Actinomycetales* (Class *Acntiomycetia*), showed a 10-fold reduction of sequences in the enriched metagenomes. At the same time, a decrease in numbers of sequences from Archaea and an increase in viruses in enrichment metagenomes suggest the impact of high-nutrient media and antibiotics on the whole microbial community and its dynamics is significant. Nutrient enrichment in the selective medium and the use of antibiotics could account for the changes observed in archaeal populations, while the increase in antibiotic-resistant bacterial strains could result in an increase in viral populations, which could be implementing a “kill the winner” strategy to control microbial dynamics (see below). Changes in the microbial composition caused by enrichments are even more evident when all metagenomes are compared. Other microbial consortia experiments have shown differential effects in bacterial and archaeal communities, for example, when salinity in extreme alkaline and saline soils was artificially reduced at the Texcoco lake [59], or the multiple recent examples in gut microbiome experiments which show significant changes in gut microbial communities when different nutrient inputs are used [60,61,62].

This microbial community analysis also highlights the different results that can be obtained when comparing metagenomics and 16S rRNA, since none of the two year sampling campaigns agree with a 2019 study describing the microbial community in El Palmar sediments, emphasizing the need for metagenomic studies to better understand the diversity of the microorganisms whose genomes are influencing the microbial processes in the environment [31,63].

Microbial communities in these sediments show that the main metabolic functions remain generally static throughout the two-year period and also after 30 days of having a mesocosms in the laboratory. However, amending the mesocosm with nutrient-rich medium and antibiotics changed the abundances of genes in those categories. The most contrasting differences were observed when the analyses were performed on specific metabolic categories, for example, sulfur metabolism, where changes in genes involved in the assimilation of organic and inorganic sulfur and in the abundance of siderophores were observed, suggesting the changes in sulfur source related to nutrient amendment go from organic to inorganic sulfur, and accompanied by a deficiency of iron in the medium. Interestingly, the addition of nutrients and antibiotics increased the number of genes in the persistent cell category, suggesting that antibiotic resistant mechanisms are activated in these communities [64]. Furthermore, microbial population dynamics and controls seem to also change in the experimental communities as genes related to phages and prophages also increase when nutrients and antibiotics are added, supporting previous observations of antibiotic-resistant bacteria being killed by prophage induction [65].

When analyzing the potential for secondary metabolite production by the microbial communities in these sediments, a clear reduction in the number of genes related to these pathways was observed in the experimental metagenomes. The number of sequences identified as related to secondary metabolite production decreased after 30 days of incubation, which would suggest that the decrease in richness and diversity affected secondary metabolite production capabilities. However, analysis of the microbial metabolite production from the 30-day control and experimental sediments shows that the highest number of secondary metabolites was observed when nutrients and antibiotics were added to the sediments. These analyses suggest that the environmental change that accompanies the reduction of diversity when nutrients and antibiotics are added has a more significant effect on the activation of secondary metabolite pathways than does the natural interaction among the microbial community.

Microbial population interactions and their community structure are thought to be controlled by secondary metabolite production in sediment bacteria [66]. Results on metagenomic and metabolomic analyses of mangrove sediment bacterial communities suggest that the potential to bring these secondary metabolites into the laboratory is mostly determined by the cultivation strategy used to isolate members of these environmental communities, since the microbial community changed drastically when selective media was used. This first effort to pair metabolomic and metagenomic data of a microbial community in artificial growth conditions suggests that biosynthetic gene clusters identified by metagenome mining can be found as compounds produced by the community. Furthermore, access to high nutrients and the use of antibiotics creates low diversity bacterial communities that can enable antibiotic-resistant bacterial cells to grow to stationary phase and produce a higher number of diverse secondary metabolites than those observed directly from an environmental microbial community. Since these cells are resistant to the antibiotics they are exposed to, results provide evidence that community dynamics could be controlled by prophage activation under “kill the winner” viral dynamics. Further analyses on the viral sequences could confirm this hypothesis, however, this analysis was beyond the scope of this paper. It becomes highly important then, to employ a wide selection of cultivation media, including different antibiotics, to access a wider diversity of microbes capable of producing diverse and novel secondary metabolites, as it seems that it will be this selection and not the origin of the sample, which will allow the bacteria in the community to reach their secondary metabolite production potential.

## 5. Conclusions

Bioprospecting in underexplored environments for the discovery of natural products benefits greatly when multi-omics approaches are used. This study not only highlights the presence of bacteria with biosynthetic capabilities for natural product synthesis in the karstic coastal sediments from “El Palmar”, but also shows that these metabolites are being produced by the microbial community under laboratory conditions. The study also highlights the unexpected effects of nutrient enrichment and antibiotic use on microbial communities, and how culture conditions can affect the abundance and biosynthetic capabilities of the targeted bacteria.

## Figures and Tables

**Figure 1 molecules-26-07332-f001:**
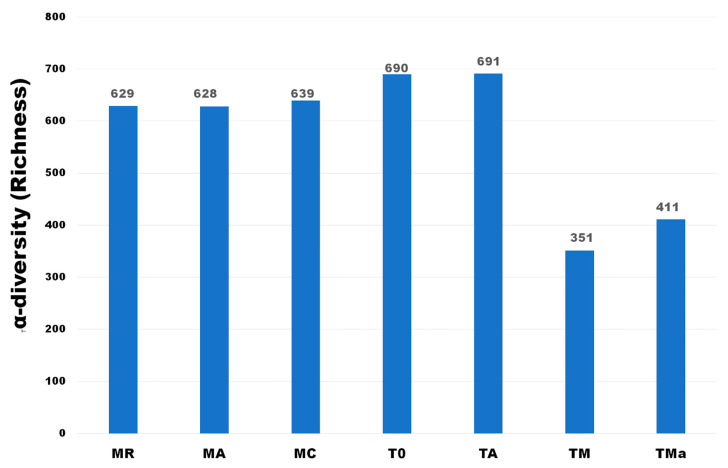
α-diversity of Phyla observed in all metagenomes analyzed. A clear decrease in richness is observed when high nutrient media and antibiotics were used in the medium. MR: sediments collected above mangrove roots. MA: sediments collected c.a. 1.5 m from mangrove roots. MC: sediments collected from running water channel with no mangrove root influence. T0: sediments from mangrove water channel (MC) sampled in 2018. TA: enrichment water control. TM and TMa: enrichment experiments without and with antibiotics (cycloheximide 10 µg/mL, gentamycin 10 µg/mL and rifampicin 1 µg/mL), respectively.

**Figure 2 molecules-26-07332-f002:**
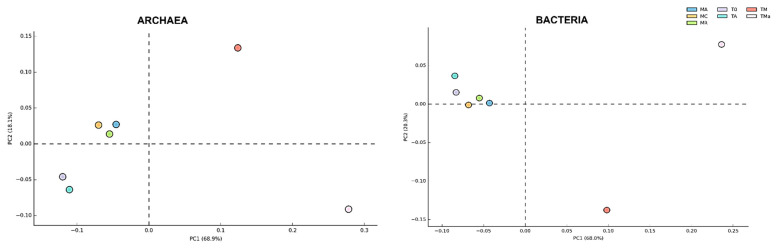
Principal component analysis (PCA) of Archaea and Bacteria Phyla in all metagenomes analyzed. While the Bacteria community does not show a broad change among communities sampled between 2017 and 2018, the Archaea community shows broad changes according to the year sampled and when nutrients and antibiotics are used in the medium. MR: sediments collected above mangrove roots. MA: sediments collected c.a. 1.5 m from mangrove roots. MC: sediments collected from running water channel with no mangrove root influence. T0: sediments from mangrove water channel (MC) sampled in 2018. TA: enrichment water control. TM and TMa: enrichment experiments without and with antibiotics (cycloheximide 10 µg/mL, gentamycin 10 µg/mL and rifampicin 1 µg/mL), respectively.

**Figure 3 molecules-26-07332-f003:**
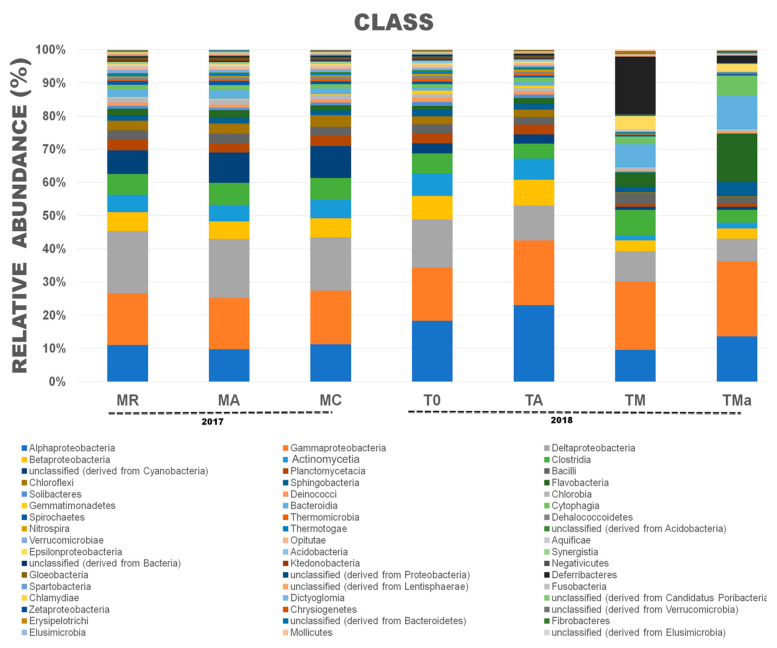
Relative abundances at the Class taxonomic level for *Bacteria* in all metagenomes analyzed. A clear reduction in the class Actinomycetia was observed when high nutrients and antibiotics were used in the medium. MR: sediments collected above mangrove roots. MA: sediments collected c.a. 1.5 m from mangrove roots. MC: sediments collected from running water channel with no mangrove root influence. T0: sediments from mangrove water channel (MC) sampled in 2018. TA: enrichment water control. TM and TMa: enrichment experiments without and with antibiotics (cycloheximide 10 µg/mL, gentamycin 10 µg/mL and rifampicin 1 µg/mL), respectively.

**Figure 4 molecules-26-07332-f004:**
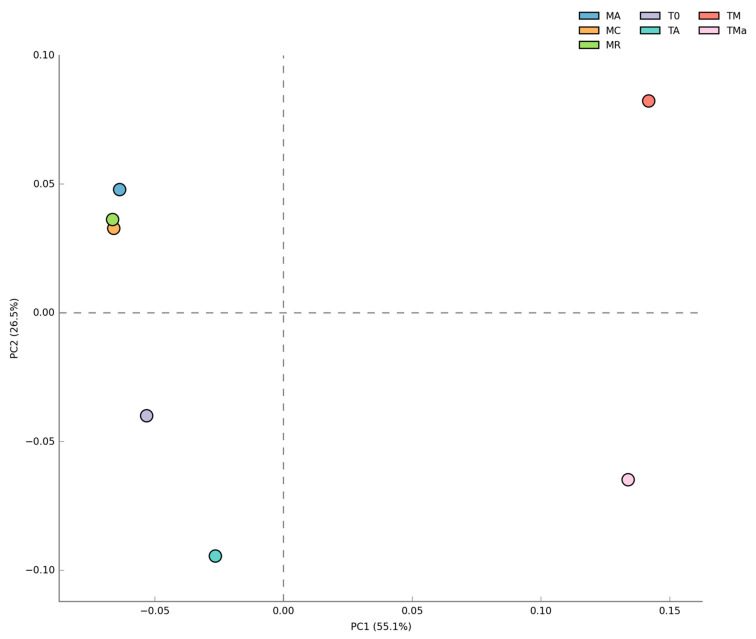
Principal component analysis (PCA) of Bacteria at the Class level for all metagenomes analyzed. No difference was observed between the two years sampled but significant differences (*p* < 0.05) were observed when high nutrients and antibiotics were used in the medium. MR: sediments collected above mangrove roots. MA: sediments collected c.a. 1.5 m from mangrove roots. MC: sediments collected from running water channel with no mangrove root influence. T0: sediments from mangrove water channel (MC) sampled in 2018. TA: enrichment water control. TM and TMa: enrichment experiments without and with antibiotics (cycloheximide 10 µg/mL, gentamycin 10 µg/mL and rifampicin 1 µg/mL), respectively.

**Figure 5 molecules-26-07332-f005:**
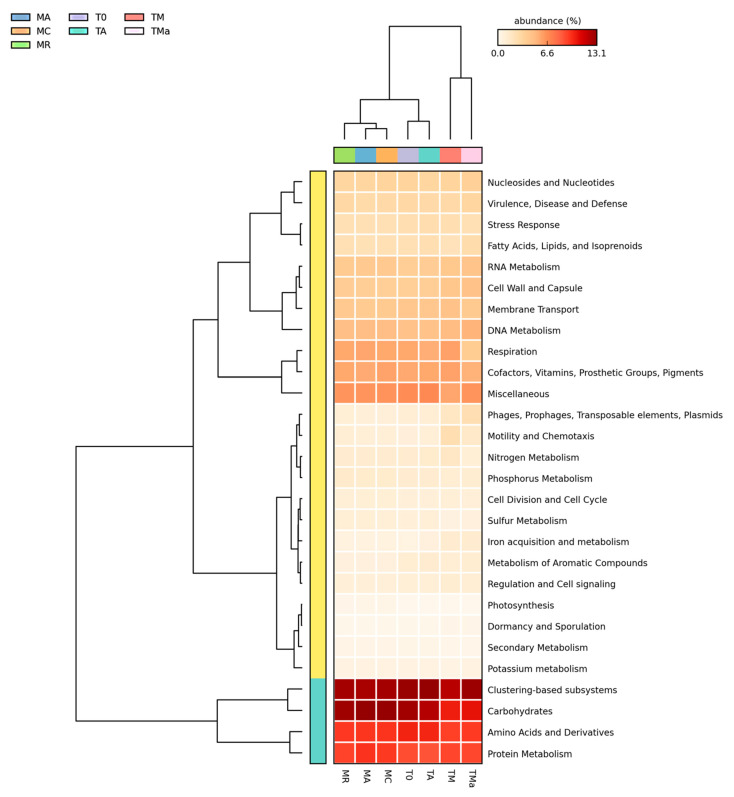
Relative abundance of primary metabolisms (MG-RAST) for all metagenomes analyzed. Most of the primary metabolism genes remain the same among all metagenomes, except some changes in phage, motility, and iron acquisition metabolisms. MR: sediments collected above mangrove roots. MA: sediments collected c.a. 1.5 m from mangrove roots. MC: sediments collected from running water channel with no mangrove root influence. T0: sediments from mangrove water channel (MC) sampled in 2018. TA: enrichment water control. TM and TMa: enrichment experiments without and with antibiotics (cycloheximide 10 µg/mL, gentamycin 10 µg/mL and rifampicin 1 µg/mL), respectively.

**Figure 6 molecules-26-07332-f006:**
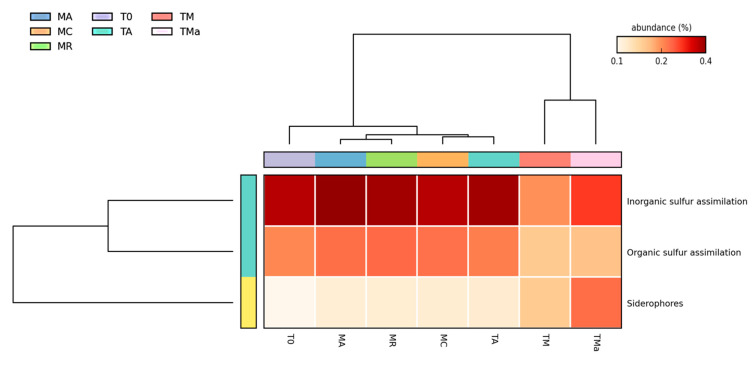
Heat map for genes involved in sulfur metabolism. Significant differences (*p* < 0.05) are observed on the abundance of organic and inorganic S assimilation and siderophores when high nutrients and antibiotics are used to amend medium.MR: sediments collected above mangrove roots. MA: sediments collected c.a. 1.5 m from mangrove roots. MC: sediments collected from running water channel with no mangrove root influence. T0: sediments from mangrove water channel (MC) sampled in 2018. TA: enrichment water control. TM and TMa: enrichment experiments without and with antibiotics (cycloheximide 10 µg/mL, gentamycin 10 µg/mL and rifampicin 1 µg/mL), respectively.

**Figure 7 molecules-26-07332-f007:**
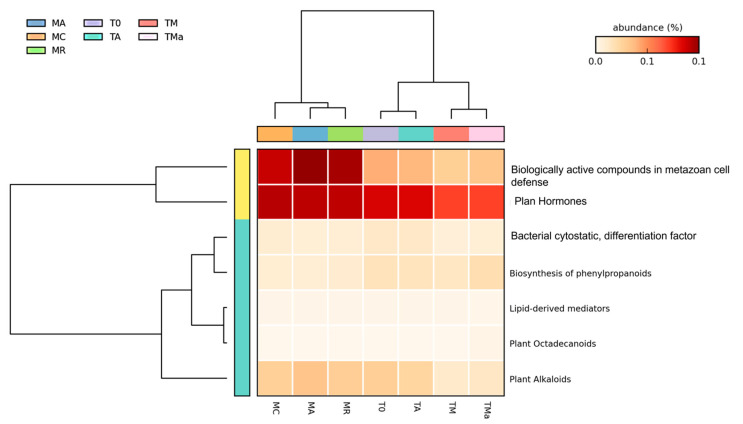
Heat map for secondary metabolism (MG-RAST) of all metagenomes analyzed. A clear difference was observed between the first and second years sampled and between the environmental and experimental metagenomes. MR: sediments collected above mangrove roots. MA: sediments collected c.a. 1.5 m from mangrove roots. MC: sediments collected from running water channel with no mangrove root influence. T0: sediments from mangrove water channel (MC) sampled in 2018. TA: enrichment water control. TM and TMa: enrichment experiments without and with antibiotics (cycloheximide 10 µg/mL, gentamycin 10 µg/mL and rifampicin 1 µg/mL), respectively.

**Figure 8 molecules-26-07332-f008:**
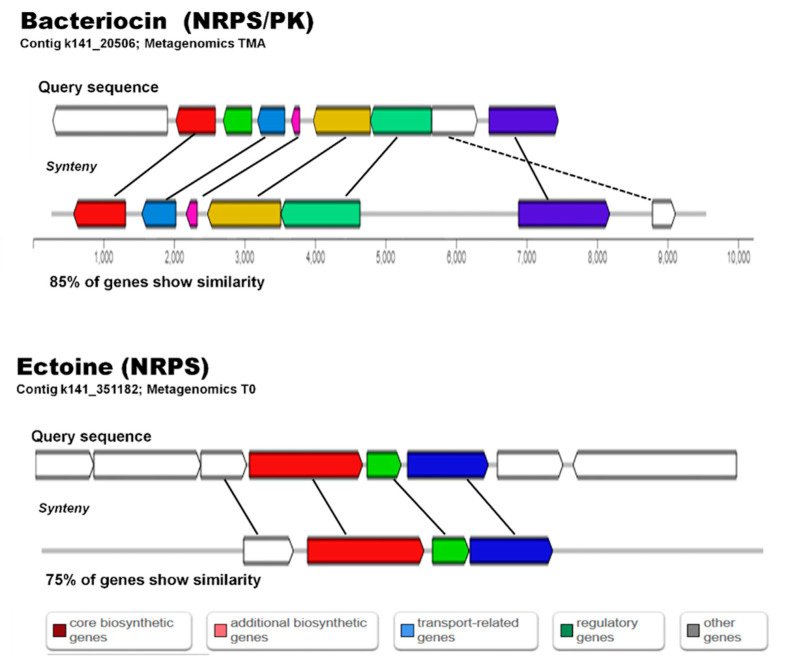
Alignment of matching sequences to biosynthetic gene clusters reported for Bacteriocin and Ectoine biosynthesis in antiSMASH.

**Figure 9 molecules-26-07332-f009:**
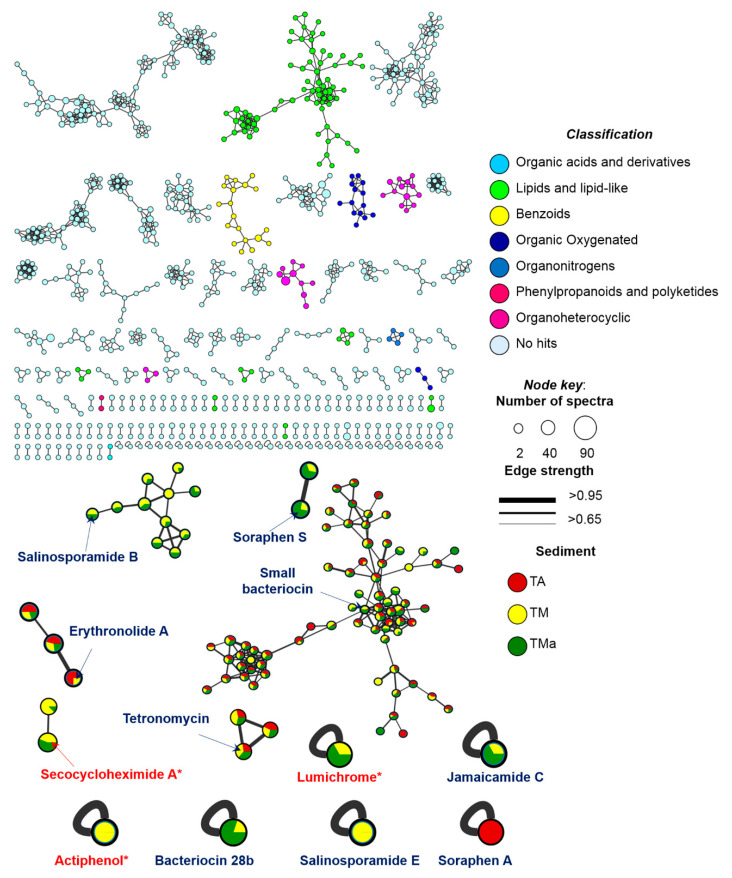
GNPS MN of communities grown under different enrichment treatments colored by super classes (legend). Arrows point the compounds annotated by GNPS (red) or by untargeted metabolomics (blue).

**Figure 10 molecules-26-07332-f010:**
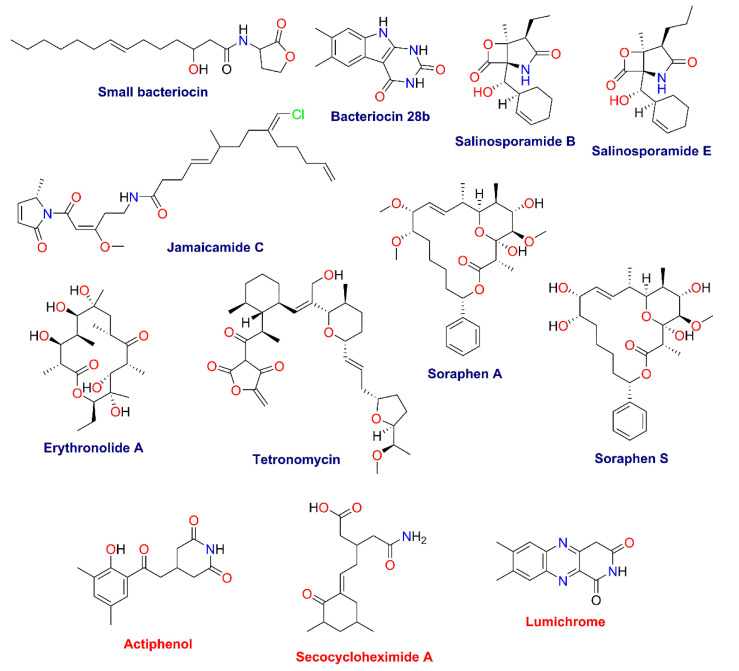
Structures of specialized metabolites found in the GNPS (red) and manual dereplication (blue) analyses.

**Table 1 molecules-26-07332-t001:** MG-RAST protein prediction.

Metagenome	Number of Sequences (Paired-End Reads)	Ribosomal RNA Reads (%)	Annotated Proteins Reads (%)	Unknown Proteins Reads (%)
MR	48,543,608	86,365 (0.124)	16,011,918 (32.98)	32,445,325 (66.84)
MA	39,014,588	1,257,879 (3.22)	10,596,346 (27.16)	27,160,363 (69.62)
MC	112,902,344	2,367,738 (2.10)	31,852,051 (28.21)	78,682,555 (69.69)
T0	22,307,461	271,350 (1.22)	9,263,336 (41.53)	12,772,775 (57.26)
TA	8,269,135	8,143 (0.10)	3,721,287 (45)	4,539,705 (54.90)
TM	6,379,606	22,005 (0.34)	4,546,110 (71.26)	1,811,491 (28.40)
TMa	8,065,138	23,003 (0.29)	5,876,101 (72.86)	2,166,034 (26.86)

**Table 2 molecules-26-07332-t002:** Sequences recovered by antiSMASH and BLAST from the assembled metagenomes.

Metagenomes	A Domain (NRPS)	C Domain (NRPS)	KS Domain (PKS-I)	Total BLAST Domains	AntiSMASH
MR	136	12	17	165	85
MA	250	13	29	292	335
MC	235	15	27	277	224
T0	85	26	29	140	163
TA	76	27	12	115	63
TM	43	27	23	93	118
TMa	15	8	14	37	63
Total	840	120	151	1122	1051

MR: sediments collected above mangrove roots. MA: sediments collected c.a. 1.5 m from mangrove roots. MC: sediments collected from running water channel with no mangrove root influence. T0: sediments from mangrove water channel (MC) sampled in 2018. TA: enrichment water control. TM and TMa: enrichment experiments without and with antibiotics (cycloheximide 10 µg/mL, gentamycin 10 µg/mL and rifampicin 1 µg/mL), respectively.

**Table 3 molecules-26-07332-t003:** Chemical annotation by GNPS and manual dereplication of sediment samples in the MN.

Compound	Molecular Formula	Exact Mass ^a^	Observed Ion (Adduct) ^b^		Mass Accuracy (ppm)	Samples
Small bacteriocin	C_18_H_31_NO_4_	325.2253	326.232 348.214	[M + H]^+^ [M + Na]^+^	−1.8 −3.1	TMa, TM, TA
Bacteriocin 28b	C_12_H_11_N_3_O_2_	229.0851	228.078	[M − H]^−^	0.7	TMa, TM
Salinosporamide B	C_15_H_21_NO_4_	279.1471	280.154 302.136	[M + H]^+^ [M + Na]^+^	−1.2 −2.7	TMa, TM
Salinosporamide E	C_16_H_23_NO_4_	293.1627	294.17	[M + H]^+^	0.1	TM
Erythronolide A	C_21_H_38_O_8_	418.2567	441.248	[M + Na]^+^	3.5	TM, TA
Jamaicamide C	C_27_H_39_ClN_2_O_4_	490.2598	491.266	[M + H]^+^	2.3	TMa, TM
Soraphen A	C_28_H_42_O_8_	506.2880	507.296 529.278	[M + H]^+^ [M + Na]^+^	1.5 0.5	TA
Soraphen S	C_27_H_40_O_8_	492.2723	491.266 493.281 515.263	[M − H]^−^ [M + H]^+^ [M + Na]^+^	2.0 2.8 1.8	TMa, TM, TA
Tetronomycin	C_34_H_50_O_8_	586.3506	587.359 609.341	[M + H]^+^ [M + Na]^+^	2.0 1.1	TMa, TM, TA
Actiphenol ^c^	C_15_H_17_NO_4_	275.1158	274.109	[M − H]^−^	3.6	TM
Secocycloheximide A ^c^	C_15_H_23_NO_4_	281.1627	280.155	[M − H]^−^	0.1	TMa, TM
Lumichrome ^c^	C_12_H_10_N_4_O_2_	242.0804	243.088	[M + H]^+^	0.1	TMa, TM

^a^ HRMS data from peaks in the LC-MS analysis of samples; ^b^ Values taken from GNPS MN; ^c^ Annotated by GNPS.

## Data Availability

All data has been made public in the MG-RAST platform (https://www.mg-rast.org; accessed on 29 September 2021) with the ID numbers presented in the manuscript.

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
