# Peer review of "A Meta-Omics Analysis Unveils the Shift in Microbial Community Structures and Metabolomics Profiles in Mangrove Sediments Treated with a Selective Actinobacterial Isolation Procedure"

_molecules, 2021, doi:10.3390/molecules26237332_

Round 1

Reviewer 1 Report

Marfil-Santana et al. aimed to profile microbial community structures and metabolites in sediment samples collected from a mangrove ecosystem using a set of meta-omics approaches and bioinformatics tools. The authors found the shift in microbial communities affected by the selective isolation procedure typically employed to target actinobacteria, a well-known source of valuable secondary metabolites. The research ideas and findings are interesting and would be attractive for Molecules' readers. However, the presentation quality, scientific interpretation, rational sense and flow, and experimental design are not well prepared/addressed and need significant improvement before further consideration. As the manuscript contains several errors that I can't list all here, the authors should thoroughly check their writing and get a critical proofread by a native English who works in the relevant field. Below you can find my comments and suggestions.

The authors do not need to put everything into the title but try to concise it instead, for example, "A meta-omics analysis unveils the shift in microbial community structures and metabolomics profiles in mangrove sediments treated with a selectively actinobacterial isolation procedure."

The abstract was poorly prepared and contained several vague statements. For example, "due to their karstic origin (line 11) - what did you mean by "their" here?" Is the first sentence essential? What did you mean by "these sediments (line 12)"? What did you mean by "their trading chemistry for communication can be targeted for secondary metabolite research (lines 12-13)"? Your research hypotheses are unclear to the readers; what exactly did you want to do? Why did you use sediment samples from different locations? Why did you compare the natural one with additional treatments (with and without nutrition supplements and/or antibiotics)? "the sediments in four locations (line 20) - how do the readers know where they were from?" Indeed, the whole sentence (lines 19-22) is simply incorrect. As I saw the results, only one sample collected in 2018 was used to compare with the other three collected in 2017, and obviously, the samples collected at different times didn't give the same results. The decrease of actinobacterial richness might not only be due to the addition of a selective medium (which medium, didn't specify) and antibiotics (which antibiotics, didn't specify) but also the way that you autoclaved the mangrove surface water. "medium also enhanced production of secondary metabolites (line 23)," only the medium - what about antibiotics? Also, you said "an unexpected decrease of actinobacteria (lines 21-22)" was observed, but in the following sentence, you said medium also enhanced production of secondary metabolites - could you explain why this happened? The title and abstract should not contain undefined abbreviations (PKS, NRPS, and GNPS). The abstract should be understandable when it stands alone without text. "These results suggest that microbial communities from "El Palmar" have biotechnological potential and support the use of diverse selective media to access a wider diversity of actinobacteria in the sediments and improve the recovery of bioactive secondary metabolites." How could you conclude this - you only used one medium? And, you reported that the medium added reduced the actinobacterial richness. The authors should elucidate the genetic diversity of actinobacteria from the metagenomics data and compare it with that observed in different treatments conducted in their study.

The introduction needs significant improvement for a better sense and flow. For example, the first paragraph should better focus on the mangrove ecosystem and its characteristics - the authors should explain why this ecosystem is the target for their study. The ecological impact of bacteria dwelling in this ecosystem should be placed at the end of the paragraph to a better link to the next one or move it to the next paragraph where you can focus solely on the mangrove forest-associated microbes.   

Tables and Figures should not contain undefined abbreviations, as they should be understandable when they stand alone without text. What do you mean by "paired read"? 

MC: sediments outside of any direct vegetation influence - where? 
MA: sediments collected 1.5 m from mangrove roots - where? 
MR: sediments collected on top of mangrove roots - where? 
T0: MC sampled on 2018. (sampled in, not on) - exact the same locations, where?
TA: experiment control - what? 
TM and TMa: enrichment experiments without and with antibiotics - what?
After reading all of these abbreviations' definitions, I still don't understand what they are.

Figure 3, Are you sure that all of the taxa are in the class category? The term Actinobacteria is the phylum, not the class. The authors should update the taxonomic classification of the domain bacteria, following Bergey's manual of systematic bacteriology or LPSN (https://lpsn.dsmz.de/domain). Please also check the spelling of the scientific names of bacteria throughout the manuscript. 

The authors should include a sampling map in their manuscript.

Author Response

Please find attached our response to Reviewer 1

Review letter Response.

Molecules Manuscript ID molecules-1431585

Reviewer 1

The authors do not need to put everything into the title but try to concise it instead, for example, "A meta-omics analysis unveils the shift in microbial community structures and metabolomics profiles in mangrove sediments treated with a selectively actinobacterial isolation procedure."

We greatly appreciate the suggestion for a new title and have decided to use the provided one by Reviewer 1 since we believe it adequately represents and highlights the study findings.

The abstract was poorly prepared and contained several vague statements. For example, "due to their karstic origin (line 11) - what did you mean by "their" here?" Is the first sentence essential? What did you mean by "these sediments (line 12)"? What did you mean by "their trading chemistry for communication can be targeted for secondary metabolite research (lines 12-13)"?

We appreciate the time taken to carefully review the manuscript. The entire manuscript was revised and each of the sections include corrections (marked in red) to allow a better understanding of the study and its findings. Also, a native-English speaker has reviewed the manuscript grammar and writing (see acknowledgements).

 Your research hypotheses are unclear to the readers; what exactly did you want to do? Why did you use sediment samples from different locations? Why did you compare the natural one with additional treatments (with and without nutrition supplements and/or antibiotics)? "the sediments in four locations (line 20) - how do the readers know where they were from?" Indeed, the whole sentence (lines 19-22) is simply incorrect

Rewritings in the Abstract, Introduction Results and Methods sections were performed in order to address these questions: Lines 20-24. Lines 100-110. Lines 111-114.

As I saw the results, only one sample collected in 2018 was used to compare with the other three collected in 2017, and obviously, the samples collected at different times didn't give the same results.

Thank you for pointing out this observation. To our surprise, samples taken from the mangrove channel (MC) almost two years apart did not show a significant change in the bacterial community structure. We had address this in the results (Lines 221-222), however, changes were observed in the archaeal community structure (Lines 178-180, figure 2).  These observations were discussed in Lines 555-564.

The decrease of actinobacterial richness might not only be due to the addition of a selective medium (which medium, didn't specify) and antibiotics (which antibiotics, didn't specify) but also the way that you autoclaved the mangrove surface water.

We appreciate the reviewer’s comment; all experimental conditions were addressed in the Materials and Methods section (Lines 425-429). However, we have added information on the section’s subtitle to further clarify (Lines 420-421). 

Regarding the autoclaved water, our experiment control (TA) was used to account for any of these differences, however, no difference was observed between the microbial community in the environmental sample (T0, MC from 2018) and the control TA, as was stated in the results section, however, wording of the sentence in this section has been modified to further clarify this point (Lines 224-226). And a line has been added in the discussion in Line 469.

"medium also enhanced production of secondary metabolites (line 23)," only the medium - what about antibiotics?

We appreciate this comment highlighting a writing mistake in the abstract. We have now added the phrase “antibiotic-amended medium” to the abstract in an effort to highlight that these observations refer to the TMa metagenome (Lines 29-32).

Also, you said "an unexpected decrease of actinobacteria (lines 21-22)" was observed, but in the following sentence, you said medium also enhanced production of secondary metabolites - could you explain why this happened?

Thank you for pointing out that this was not cleared enough in the discussion section (Lines 637-640). We mentioned it in the abstract in hopes of capturing the readers’ curiosity, however, we have now added a line in the abstract to clarify the result (Lines 34-36).

The title and abstract should not contain undefined abbreviations (PKS, NRPS, and GNPS). The abstract should be understandable when it stands alone without text.

Thank you for this observation, the title and abstract have been revised and re-written.

These results suggest that microbial communities from "El Palmar" have biotechnological potential and support the use of diverse selective media to access a wider diversity of actinobacteria in the sediments and improve the recovery of bioactive secondary metabolites." How could you conclude this - you only used one medium? And, you reported that the medium added reduced the actinobacterial richness.

Thank you for highlighting this phrase. We believe that our metagenomic mining and metabolomic results (as well as results from a parallel cultivation-dependent experiment not shown in this manuscript, yet mentioned in Line 112), are strong enough evidence to support our first statement on the biotechnological potential. Also, we have shown that the use of only one selective-medium reduced, and not increased, the abundance of actinomycetes in the mangrove sediment samples, therefore, we used this result to recommend the use of multiple media instead of trusting only one for the selective isolation of any bacterial group.

The authors should elucidate the genetic diversity of actinobacteria from the metagenomics data and compare it with that observed in different treatments conducted in their study.

Thank you for pointing out that our discussion on the change of actinobacteria was not strong enough in figure 3 and Lines 205-208. To clarify this we have added further taxonomic resolution comparisons in figure 3. 

The introduction needs significant improvement for a better sense and flow. For example, the first paragraph should better focus on the mangrove ecosystem and its characteristics - the authors should explain why this ecosystem is the target for their study. The ecological impact of bacteria dwelling in this ecosystem should be placed at the end of the paragraph to a better link to the next one or move it to the next paragraph where you can focus solely on the mangrove forest-associated microbes.  

Thank you for carefully reading the manuscript. We have rewritten the introduction and other sections of the manuscript to improve the sense and flow. Also, another native English speaker colleague has performed a revision of the Grammar and English writing.

Tables and Figures should not contain undefined abbreviations, as they should be understandable when they stand alone without text. What do you mean by "paired read"? 

Thank you for suggesting this. All figures and tables have been revised to avoid this. The word “paired-end” has now been added to Table 1.

MC: sediments outside of any direct vegetation influence - where? 

MA: sediments collected 1.5 m from mangrove roots - where? 

MR: sediments collected on top of mangrove roots - where? 

T0: MC sampled on 2018. (sampled in, not on) - exact the same locations, where?

TA: experiment control - what? 

TM and TMa: enrichment experiments without and with antibiotics - what?

After reading all of these abbreviations' definitions, I still don't understand what they are.

Thank you for highlighting that there was a paragraph missing in the results section. All definitions have been expanded in this section under the subtitle “Sampling location, collection dates and experimental treatments” and they have been edited in all figures.

Figure 3, Are you sure that all of the taxa are in the class category? The term Actinobacteria is the phylum, not the class. The authors should update the taxonomic classification of the domain bacteria, following Bergey's manual of systematic bacteriology or LPSN (https://lpsn.dsmz.de/domain). Please also check the spelling of the scientific names of bacteria throughout the manuscript.

Thank you for highlighting the update on the class Actinomycetia, the correction has been made throughout the document.

The authors should include a sampling map in their manuscript.

Thank you for this suggestion. A map of the sampling location has been added

Reviewer 2 Report

Line 10 to 19: The introduction section within the abstract seems too lengthy. Better to reduce the basic info and focus on the findings, significance, and implications.

Line 21-22: I feel that the sentence should be rephrased to make it easily readable.

Line 26: Please expand “GNPS,” same goes for all abbreviations throughout the MS

Line 24-27: is the presence of PKS and NRPS biosynthetic genes enough to claim that it has biotechnological potential? Metagenomes from other ecosystems also have similar reports. What is the novelty of this study?

Line 87: Please expand “MA, MR and MC”

Line 85: Samples were obtained in 2017 and 2018. This would introduce batch effects. Were any corrections done to eliminate the batch effect?

Line 95: As per my understanding, only one sample before and one after antibiotic treatment was used for analysis. Is it sufficient to draw a conclusion based on just one sample per treatment?

Line 140-141: A brief explanation of the higher abidance of viral reads in the discussion section would be interesting.

Line 156: The caption describes T0 and MC is a bit confusing. Would you mind writing separately?

Line 160: Please mention the p values for “significant”

Line 187: Sentence could be rephrased.

Line 186-194: PCA was done for Phyla and Class. What necessitates the need for PCA for both taxonomic ranks? Would you mind mentioning it in the MS?

Line 213: please re-verify the database name: “M5g database”

Line 360-366: Why the need for a different font? The same applies to other parts of the MS.

Line 472: Please cite evidence that karstic conditions could be a source of novel natural products.

Line 514: As per my understanding, the metagenomics and 16S rRNA data showed different results. Is it because of a difference in the technique or difference in database/pipelines? Have the authors checked the similarity/difference between 16 rRNA data and metagenomics data annotated with the same database, e.g., RDP, Greengenes, etc., used for 16s rRNA?

Conclusion: Conclusion seems to be missing in the MS. Please add.

The results and discussion lack descriptions of the metabolisms and PKS, especially regarding the study's novel findings.

The overall MS requires further corrections throughout the MS for language and description of the novelty.

Author Response

Please find attached our response to reviewer 2

Review letter Response.

Molecules Manuscript ID molecules-1431585

Reviewer 2

Line 10 to 19: The introduction section within the abstract seems too lengthy. Better to reduce the basic info and focus on the findings, significance, and implications.

Thank you for taking the time to carefully review the manuscript. The abstract and introduction have been rewritten and the findings, significance, and implications have been highlighted.

Line 21-22: I feel that the sentence should be rephrased to make it easily readable.

Line 26: Please expand “GNPS,” same goes for all abbreviations throughout the MS

Abstract has been rewritten, all abbreviations have been spelled out in abstract and at first mention in manuscript.

Line 24-27: is the presence of PKS and NRPS biosynthetic genes enough to claim that it has biotechnological potential? Metagenomes from other ecosystems also have similar reports. What is the novelty of this study?

We agree with the reviewer that many other ecosystems also have similar reports, however, this characteristic sediment environment (karst ecosystem influenced by underground river water and seawater) has not been explored for natural product discovery. There has only been one study cited in line 79, reference 30 (Martínez-Nuñez, 2020) as the only previous exploration of this site using NRPS amplicon sequencing from nearby sediments. However, this previous study didn’t focus on natural product discovery and only described the diversity of this set of genes found in their sample. We believe that the metabolomics study performed on the sediments shows strong evidence that some of the compounds mined from the metagenome were found being produced by the microbial community under laboratory conditions, and therefore, the biotechnological potential of the microbial community is real. Also, we performed parallel cultivation-dependent studies (data not shown) which have shown that the bacteria isolated from these sediments produce bioactive secondary metabolites, this information has been added to the manuscript to support our conclusion (Lines 112-113).

Line 87: Please expand “MA, MR and MC”

Done, missing segment included in the Results section Lines 101-117

Line 85: Samples were obtained in 2017 and 2018. This would introduce batch effects. Were any corrections done to eliminate the batch effect?

Thank you for pointing this out. No, no batch effect corrections were performed. Both samples were sent to the same laboratory for sequencing. We did however, correct for differences in sequencing depth and normalized data in all samples to the least number of sequences obtained (Tm with c.a. 8 million reads). Although no great difference was observed between most of the community from 2017 and 2018 (revised figure 2 with Phyla PCoA), we did observe a significant difference between four phyla (shown in supplementary figure 7), and this difference has now been addressed in the discussion section suggesting environmental conditions or a batch effect could be causing it (Lines 613-616). Also, the difference at the phylum level was not observed at the family level within the order Actinomycetales (where actinomycetes would be classified), therefore, our analyses on the selective media should still be valid (see supplementary figure 11).

Line 95: As per my understanding, only one sample before and one after antibiotic treatment was used for analysis. Is it sufficient to draw a conclusion based on just one sample per treatment?

Thank you for pointing out how missing information made this confusing. One site inside the location sampled (mangrove channel MC) was re sampled for the laboratory experiment. There were three cores from this site taken to the laboratory and the experiments were performed by triplicate. DNA was then extracted from the triplicates and pooled together for sequencing.  This missing information has now been added to the experimental section.

Line 140-141: A brief explanation of the higher abidance of viral reads in the discussion section would be interesting.

A brief hypothesis of this phenomenon had been included in the discussion section, Lines 615-619 and was incorporated further in Lines 642-646.

Line 156: The caption describes T0 and MC is a bit confusing. Would you mind writing separately?

Captions in figures have been corrected and a specific paragraph has been added in the Results section Lines 101-117

Line 160: Please mention the p values for “significant”

Significance was mentioned in supplementary figures and refers to a p value of p<0.05 it has now been added to figures and text in MS.

Line 187: Sentence could be rephrased.

Manuscript writing has been revised

Line 186-194: PCA was done for Phyla and Class. What necessitates the need for PCA for both taxonomic ranks? Would you mind mentioning it in the MS?

Thank you for pointing out that Figures 2 and 4 do not show clearly that one is for the Archea and another one for Bacteria. A higher taxonomic level was used for Bacteria since we were interested in the Class level changes to monitor differences within the Actinomycetia (previously class Actinobacteria). We believe the manuscript rewriting makes this difference more clear.

Line 213: please re-verify the database name: “M5g database”

Done, M5rna was used. Line 279

Line 360-366: Why the need for a different font? The same applies to other parts of the MS.

We apologize for the mistake in the formatting, it has now been corrected.

Line 472: Please cite evidence that karstic conditions could be a source of novel natural products.

We have stated why we believe this unique environment can be a source of novel natural products and added a reference to Line 594.

Line 514: As per my understanding, the metagenomics and 16S rRNA data showed different results. Is it because of a difference in the technique or difference in database/pipelines? Have the authors checked the similarity/difference between 16 rRNA data and metagenomics data annotated with the same database, e.g., RDP, Greengenes, etc., used for 16s rRNA?

Thank you for suggesting this analysis, however, the data between a metagenomic study and a 16S rRNA amplicon sequencing study cannot be fully compared and can only be used to observe differences caused by the technique since 16rRNA amplicon studies use sequences covering a specific region of the 16S rRNA gene, while metagenomic studies use sequences that have been randomly amplified. Therefore, even the 16S rRNA sequences from the metagenomics study can’t be used for a direct comparison with the previous study, as the amplified regions most probably differ amongst them.

Conclusion: Conclusion seems to be missing in the MS. Please add.

A conclusion has been added.

The results and discussion lack descriptions of the metabolisms and PKS, especially regarding the study's novel findings.

The overall MS requires further corrections throughout the MS for language and description of the novelty.

The manuscript has been rewritten to highlight the novel findings.

Reviewer 3 Report

This paper covers many issues related to the changes in the microbial communities from coastal karstic mangrove sediments. The article is very interesting. The work is well prepared, The design of the experiments is appropriate and clearly described in the text,  results clearly confirmed the hypothesis.

The main problem I had when reviewing this work is the lack of attached supplementary materials: Figures S 1 - S 19 and Tables S1 - S 6. The lack of these materials made it difficult to analyze the results. 

Authors should correct some typos e.g. line 119 "figure s 2 A and B"; line 227 "table S 3".

Author Response

Please find attached our response to reviewer 3

Review letter Response.

Molecules Manuscript ID molecules-1431585

Reviewer 3

This paper covers many issues related to the changes in the microbial communities from coastal karstic mangrove sediments. The article is very interesting. The work is well prepared, the design of the experiments is appropriate and clearly described in the text, results clearly confirmed the hypothesis.

Thank you for your comments. We agree that the findings are interesting and should be published. Also, following other reviewers’ suggestions, we have rewritten parts of the manuscript to make our points easier to come across.

The main problem I had when reviewing this work is the lack of attached supplementary materials: Figures S 1 - S 19 and Tables S1 - S 6. The lack of these materials made it difficult to analyze the results. 

Our sincere apologies, we believe there must have been a mistake in the upload. We will make sure all supplementary material is uploaded correctly and available to all readers.

Authors should correct some typos e.g. line 119 "figure s 2 A and B"; line 227 "table S 3".

Thank you, we have corrected the typos.

Round 2

Reviewer 1 Report

The manuscript has improved substantially only in the part of the abstract. A thorough revision in the other sections has been required, especially for the English usage mentioned previously. The use of abbreviations is still a problem often found everywhere. For example, GNPS is unnecessary for the abstract, and MC is redundant in paragraph lines 95-112 and most of the figures' legends. Try to provide the abbreviation in a general way, for instance, mangrove rhizosphere sediments (MR), sediments located 1.5 m away from mangrove roots (MA), and sediments located near water channels and not associated with mangrove roots (MC). Errors in English usage and inconsistent units (L/l, mL/ml) have been found throughout the manuscript. Every figure needs significant improvement for the resolution, axis titles, and legends. In figure 3, why do some have the term "(class)," why do some have not? Principal coordinates analysis (PCoA) and principal component analysis (PCA) are not the same; the authors might confuse the two methods indicated in the text and the figure legend. 

Author Response

Reply to Reviewer 1 MDPI

The manuscript has improved substantially only in the part of the abstract. A thorough revision in the other sections has been required, especially for the English usage mentioned previously.

Thank you for your suggestion. A thorough revision of the English has been performed, the edits on the first round was highlighted in red, the new revision highlights is presented in yellow. Manuscript has been now revised by Dr. J. Winter and Dr. E. Gontang.

The use of abbreviations is still a problem often found everywhere. For example, GNPS is unnecessary for the abstract, and MC is redundant in paragraph lines 95-112 and most of the figures' legends. Try to provide the abbreviation in a general way, for instance, mangrove rhizosphere sediments (MR), sediments located 1.5 m away from mangrove roots (MA), and sediments located near water channels and not associated with mangrove roots (MC).

A clearer description of the sediments collected has been provided (Lines 113-118 and figure legends). Since Reviewers asked for a figure legend which could allow the figure to stand alone and be understood, we decided to explain clearly in figure legends what all abbreviations mean.

Errors in English usage and inconsistent units (L/l, mL/ml) have been found throughout the manuscript.

Thank you for your comments. English has been corrected by two more native English speakers who have been acknowledged (red and yellow edits in manuscript).  Units have been revised.

Every figure needs significant improvement for the resolution, axis titles, and legends. In figure 3, why do some have the term "(class)," why do some have not?

Thank you for your comment. Figures in paper are low resolution for revision purposes only. However, another two zip files (one for figures and one for supplementary figures) with resolutions of 300 dpi and higher were also provided during the process.

The names according to the figure originally created by the MG RAST platform with the analysis had been kept before, however, figure 3 has now been corrected.

Principal coordinates analysis (PCoA) and principal component analysis (PCA) are not the same; the authors might confuse the two methods indicated in the text and the figure legend. 

Thank you for your observation, the mistake has been corrected and PCA has been used throughout the manuscript referring to the correct analysis performed.

Reviewer 2 Report

No Comments

Author Response

We appreciate Reviewer 2's comments throughout the review process as we recognize the time and effort this task takes. We have edited multiple sections of the paper to make sure the importance of our work and findings is highlighted.